# A Novel Cold-Adapted and High-Alkaline Alginate Lyase with Potential for Alginate Oligosaccharides Preparation

**DOI:** 10.3390/molecules28176190

**Published:** 2023-08-22

**Authors:** Hai-Ying Wang, Zhi-Fang Chen, Zhi-Hong Zheng, Hui-Wen Lei, Hai-Hua Cong, Hai-Xiang Zhou

**Affiliations:** 1Key Laboratory of Sustainable Development of Polar Fishery, Ministry of Agriculture and Rural Affairs, Yellow Sea Fisheries Research Institute, Chinese Academy of Fishery Sciences, Qingdao 266071, China; wanghy@ysfri.ac.cn (H.-Y.W.); czf981115@163.com (Z.-F.C.); m15232177216@163.com (Z.-H.Z.); leihuiwen0113@163.com (H.-W.L.); 2Shandong Peanut Research Institute, Qingdao 266100, China; 3College of Fisheries and Life Science, Dalian Ocean University, Dalian 116023, China; 4College of Food Science and Engineering, Dalian Ocean University, Dalian 116023, China; 5College of Food Science and Technology, Suzhou Polytechnic Institute of Agriculture, Suzhou 215008, China

**Keywords:** alginate lyase, cold-adapted, high-alkaline, alginate oligosaccharides, *Yarrowia lipolytica*

## Abstract

Alginate oligosaccharides (AOs) prepared through enzymatic reaction by diverse alginate lyases under relatively controllable and moderate conditions possess versatile biological activities. But widely used commercial alginate lyases are still rather rare due to their poor properties (e.g., lower activity, worse thermostability, ion tolerance, etc.). In this work, the alginate lyase Alyw208, derived from *Vibrio* sp. W2, was expressed in *Yarrowia lipolytica* of food grade and characterized in order to obtain an enzyme with excellent properties adapted to industrial requirements. Alyw208 classified into the polysaccharide lyase (PL) 7 family showed maximum activity at 35 °C and pH 10.0, indicating its cold-adapted and high-alkaline properties. Furthermore, Alyw208 preserved over 70% of the relative activity within the range of 10–55 °C, with a broader temperature range for the activity compared to other alginate-degrading enzymes with cold adaptation. Recombinant Alyw208 was significantly activated with 1.5 M NaCl to around 2.1 times relative activity. In addition, the endolytic Alyw208 was polyG-preferred, but identified as a bifunctional alginate lyase that could degrade both polyM and polyG effectively, releasing AOs with degrees of polymerization (DPs) of 2–6 and alginate monomers as the final products (that is, DPs 1–6). Alyw208 has been suggested with favorable properties to be a potent candidate for biotechnological and industrial applications.

## 1. Introduction

Macroalgae are abundant marine biomasses that play an important role in the ecosystem, containing multiple active substances, such as polysaccharides, polyphenols, plant hormones, and so on, whose extraction requires the efficient disruption of the cell wall structure [1]. As the highest amount of polysaccharide formed in the cell wall and intracellular matrix of most brown algal, alginate has been used abroad in the food processing, cosmetic, pharmaceutical, and chemical industries [2,3]. Two monomeric units, α-l-guluronate (G) and β-d-mannuronate (M), are linked through 1, 4-O-glycoside bonds to form an unbranched acidic polysaccharide, constituting three kinds of configurations: homopolymeric G block (polyG) and M block (polyM), and alternating or random MG block (polyMG) [4]. Accompanied by the destruction of the cell wall structure for acquiring its intracellular bioactive substances, macromolecular alginate is depolymerized and transformed into oligomers [5]. Moreover, alginate oligosaccharides (AOs) have received more attention thanks to their lower molecular masses, better solubility, and promising biological activities compared to alginate polymers [6]. Due to their beneficial health effects, such as anti-oxidation [7], anti-obesity [8], anti-hypertension [9], anti-tumor [10], neuroprotection [11], immunoregulation [12], and proliferating probiotics [13], AOs are comprehensively considered to be recommendable ingredients of functional foods [14]. In addition, the oligosaccharides derived from alginate also possess other activities, e.g., promoting plant growth [15], cryoprotective function [16], stabilizing the nanoparticle effect [17], etc. Thus, they are of great interest to researchers in the food and pharmaceutical industries.

AOs prepared by alginate depolymerization by enzymatic catalysis with relatively mild, green, and controllable conditions possess more desirable properties and activities compared to those prepared using physicochemical methods [18]. Alginate lyase, which cleaves the 1, 4-linkages in alginate on the basis of a β-elimination mechanism, can generate unsaturated saccharides with a double bond between C4 and C5 at the new non-reducing terminal uronic acid, called 4-deoxy-L-erythro-hex-4-enopyranosyluronic acid (usually represented using ‘Δ’) [19]. The alginate lyases are generally divided into endo- and exo-types according to their catalytic pattern [20]. The exolytic alginate lyases (EC 4.2.2.26) degrade the alginate from the end of the chain, manufacturing a homogeneous product, generally, uronic acid monomers (Δ) [21,22,23,24]. By contrast, the endolytic ones mainly cut the glycosidic linkages inside the alginate molecule to finally release a variety of unsaturated AOs [25,26,27,28]. Intriguingly, some alginate lyases that tailor alginate in an endo-type manner were also found to generate AOs along with some monosaccharides, perhaps with exolytic activity in these enzymes [29]. Among the endolytic alginate lyases, polyM-specific lyases (EC 4.2.2.3) dramatically prefer degrading polyM to polyG [30], polyG-specific lyases (EC 4.2.2.11) show far higher activity against polyM [31], and the activities of polyMG-specific lyases (EC 4.2.2.-) toward homogenous substrates can be neglected compared to that against MG-block [32]. Compared to these, the bifunctional ones (EC 4.2.2.-), with no obvious bias, can depolymerize all kinds of substrates more efficiently [33,34]. In addition, alginate lyases can be classified into many polysaccharide lyase (PL) families, including the PL5, 6, 7, 8, 14, 15, 17, 18, 31, 32, 34, 36, 39, and 41 families to date [26], whereas most alginate lyases reported belong to the PL7 family [35]. By now, alginate lyases have been utilized in a variety of fields, such as the elucidation of alginate structures [36], the preparation of brown seaweed protoplast [37], and therapy for cystic fibrosis by inhibiting pathogen bacteria proliferation by the disruption of their biofilm [38,39].

So far, a number of alginate lyases derived from mollusks [24], algae [40], viruses [41], fungi [42], and especially bacteria [43,44,45,46], have been identified, expressed, and characterized. However, commercial alginate lyases that have been used widely are still rather rare [35]. The enzymes reported to have specific characteristics are undoubtedly welcome, e.g., thermo-stability [47], pH stability [48], cold adaptation [49], alkaline or acidic adaptation [50], etc. Significantly, relatively high activities under low temperatures (<30 °C generally) make cold-adapted enzymes more available due to their biocatalysis at room temperature, which is a prerequisite to enhance the sustainability of alginate lyases, reduce their contamination risk, and sequentially cut costs in their industrial production [33]. Furthermore, the majority of alginate lyases possess their optimal pH in the range of neutral to slightly alkaline [35]. The ones that prefer working in high-alkaline conditions are few; however, relatively high-alkaline conditions are a benefit for alginate extraction and dissolution from brown seaweed, thereby improving AOs production efficiency. Several reported high-alkaline alginate lyases predominantly originated from *Chlorella* viruses or marine invertebrates [51]. As for the alginate lyases from bacteria, optimum pH values higher than 9.0 are unusual [50]. Hence, it is of extreme urgency to excavate new alginate lyases with promising properties, including both cold and high-alkaline adaptation.

Previously, a fantastic sodium alginate-degrading bacterium strain, *Vibrio* sp. W2, was isolated from abalone viscera [48]. Three alginate lyases have been discovered and researched based on genome mining [48,50,52]. In this work, another bifunctional PL7 family alginate lyase, Alyw208, with favorable high-alkaline and cold-adapted properties, was expressed using food-grade *Yarrowia lipolytica* as a host. Then, its purification and characterization were performed in detail, confirming it is suitable for the industrial preparation of AOs.

## 2. Results and Discussion

### 2.1. Bioinformatics Analysis of the Alginate Lyase Alyw208

*Vibrio* sp. W2, a marine bacterium strain isolated from abalone viscera in a previous work, can degrade agarose and alginate efficiently [48]. With the help of genome mining, several alginate lyases, including Alyw201, Alyw202, and Alyw203, have been discovered, expressed, and characterized. Interestingly, all the three enzymes belong to the PL7 family [48,50,52]. For the members of this family, the alginate-degrading activity is the only verified function, and thus putative sequences according with the common feature of the protein sequences from this family could be regarded as possible candidates of alginate lyases.

In this study, a predicted alginate lyase-coding gene, named *alyw208*, was annotated in *Vibrio* sp. W2 genome with an ORF of 1026 bp (Genbank number OR423045), encoding a protein composed of 341 residues. The putative protein Alyw208 had a supposed signal peptide (Met^1^-Gly^29^) at the N-terminus, a theoretical molecular weight (Mw) of 33.9 kDa, and an isoelectric point (pI) of 4.76. Detailed retrieval in the Conserved Domain Database (CDD) from NCBI revealed Alyw208 contained a single domain (Phe^69^-Glu^336^) of the PL7 family pertaining to the superfamily 2. Additionally, Alyw208 was subjected to a BLAST analysis in NCBI online and discovered to be evolutionarily close to the members in the PL7 family. A phylogenetic tree was constructed for Alyw208 together with other members of the PL7 family (Figure 1), which illustrated that Alyw208 was located at the same branch with the alginate lyase from *Vibrio aphrogenes* (Genbank number WP_086981030). According to the sequence alignment between Alyw208 and other enzymes in the PL7 family (Figure 2), three typical sequences of Alyw208 were identified, consistent with the conserved regions, i.e., RXELR, Q(I/V)H, and YFKAGXYXQ, of the PL7 family proteins. The sequences are responsible for the substrate-binding and catalyzing functions [44,46]. All the above information indicated that Alyw208 was a new member of the PL7 family. Furthermore, studies have reported Q(I/V)H is associated with substrate specificity. The QIH motif mainly recognizes polyG or polyMG block, and the valine residue here represents polyM predilection [33,49], implying polyG-preferential substrate specificity of Alyw208. The 3D structure of Alyw208 has been constructed by homology modeling, and the molecular graphic image was prepared using PyMOL 2.0.3 (Schrödinger, LLC, Portland, OR, USA), as displayed in Appendix A. Three catalytic residues (Q174, H176, and Y285) were predicted in Alyw208 on the basis of previous research (Figure 2 and Appendix A). Gln^174^ is responsible for acting with the C6 carboxyl group at subsite +1; His^176^ abstracts the C5 proton as a general base; and Tyr^285^ donates a proton to the O4 atom as a general acid [44].

### 2.2. Secretory Expression and Purification of Alyw208

Although *Escherichia coli* or *Pichia pastoris* have stronger expression capability, in this work, we adopted an engineering strain of food grade, *Y. lipolytica* URA^−^, for secretory expression of the alginate-degrading enzyme Alyw208. A variety of recombinant enzymes including alginate lyases were expressed in *E. coli*, but the extensive application of this expression system in industrial production was impeded owing to the poor protein secretion ability and the insecurity factors (presence of bacteria endotoxin and pyrogen) [53]. Similar problems appeared in the application of *P. pastoris*. For the sake of increased protein secretion, methanol has to be employed as an inducer during fermentation, which brings a great insecurity risk to the enzyme-producing fermentation staff [54]. By contrast, in the case of employing *Y. lipolytica* as the host with an outstanding extracellular secretion ability, there is no need to add any antibiotic or inducer, which makes this expression system an ideal choice for heterologous expression [55]. After the positive transformant was cultivated in the GPPB medium for 84 h, the extracellular alginate lyase activity reached 37.76 ± 3.05 U/mL. Furthermore, 100 mL fermentation supernatant was subject to Ni-IDA agarose affinity chromatography. The purified enzyme showed the specific activity of 729.5 U/mg, as detailed in Table 1.

The theoretical Mw of Alyw208 without a signal sequence was 33.9 kDa, similar to those (25 to 40 kDa) of the alginate lyases with a single domain of the PL7 family [33,43]. An enzyme larger than 50 kDa in this family always contained another domain, such as the carbohydrate-binding module (CBM) domain [46], the F5/8 type C domain [34], or another alginate-lyase 2 domain [49], except the alginate-lyase 2 superfamily domain [56]. The purified recombinant Alyw208 with alginate-depolymerizing activity was further analyzed by SDS-PAGE (Figure 3). As expected, a single band (nearly 40.0 kDa) appeared in the gel between the markers of 35 and 48 kDa, a little larger than the theoretical Mw of Alyw208, which was due to the fusion of a 6×His-tag. Furthermore, the discrepancy might be attributed to the diverse protein modifications introduced during the post-translational processing, among which glycosylation contributed the most [54,57]. The future research in regard to the glycosylation in recombinant Alyw208 can be performed to investigate the influences of eukaryotic modifications on the enzymatic function. A number of studies indicated that the recombinant enzymes modified by glycosylation would have better thermal stability than the native ones [54,58]. However, there were some studies reporting that glycosylation affected enzyme structures and catalytic activities [59].

### 2.3. Substrate Specificity and Kinetic Parameters of Alyw208

Three substrates (polyM, polyG, and sodium alginate) were employed to study the substrate specificity of Alyw208 (Figure 4). Alyw208 slightly preferred polyG over the other two substrates, which, however, did not cover up its bifunctionality of degrading both polyG and polyM (Figure 4). Compared with the alginate lyases strictly tailoring single polyG or polyM block [60,61], Alyw208 with broad substrate specificity was able to depolymerize alginate efficiently and make full use of the substrate, just as most bifunctional alginate-degrading enzymes [33,62]. In addition, the slight polyG preference roughly validated the prediction by bioinformatics analysis, i.e., the polyG or polyMG predilection of the QIH motif. Although PL7 family members have diverse substrate specificities (Table 2), that rule has been confirmed repeatedly. Like Alyw208, the alginate lyase A1m from *Agarivorans* sp. JAM-A1m [51], AlyS02 from *Flavobacterium* sp. S02 [33], and Aly08 from *Vibrio* sp. SY01 [31] preferred polyG as the substrate, and Algb from *Vibrio* sp. W13 exhibited the highest and the second activities against polyMG and polyG, respectively [49]. They all shared the QIH region. 

The alginate lyase A9mT from *Vibrio* sp. JAM-A9m [61] and AlyVOA and AlyVOB from *Vibrio* sp. O2 [67] with higher polyM-degrading capability contained the QVH motif. Intriguingly, exceptions occurred. For example, the QIH motif appeared in the protein sequences of FlAlyA and Aly7B_Wf, which, however, were polyM-preferred [44,68]. In addition, rAlgSV1-PL7 from *Shewanella* sp. YH1 with the QVH motif had higher activity on polyG [69].

The enzyme activity of Alyw208 was determined with sodium alginate solutions at a series of concentrations, on the basis of which the kinetic parameters of recombinant Alyw208 were estimated. A Lineweaver–Burk plot was made with the reciprocals of substrate concentration 1/[S] and reaction rate 1/V as the abscissa and ordinate, respectively (Figure 5). The *K*_m_ and *V*_max_ values of the recombinant enzyme were 2.409 mg/mL and 0.0746 μmol/min, respectively, according to the cross-intercept, vertical-intercept, and curve slope. Consequently, the *k*_cat_ (turnover number) and the *k*_cat_*/K*_m_ (catalytic efficiency constant) were computed as 154.5 min^−1^ and 64.13 mg^−1^·mL·min^−1^, respectively. Although a large number of alginate lyases have been characterized, the kinetic constants of them were rarely reported, and the reported ones were even presented in different measuring units. The *k*_cat_ of Alg7A from *Vibrio* sp. W13 toward sodium alginate was 38.98 min^−1^ [46], and the *K*_m_*, k*_cat_, and *k*_cat_*/K*_m_ values of the alginate lyase alyPG from *Corynebacterium* sp. ALY-1 were deduced as 270 mg/mL, 58.2 min^−1^, and 0.216 mg^−1^·mL·min^−1^, respectively [44]. Alyw208 exhibited lower *K*_m_ and higher *k*_cat_ and *k*_cat_*/K*_m_ than the reported alginate lyases, demonstrating stronger substrate-binding affinity and higher catalysis efficiency toward alginate.

### 2.4. Effects of Temperature on the Activity and Stability of Recombinant Alyw208

The alginate lyase activity was measured at different temperatures. As displayed in Figure 6A, recombinant Alyw208 manifested the highest activity at 35 °C, with the relative activity of over 70% shown within the temperature range of 10–55 °C, which illustrated this enzyme was cold-adapted. Cold-adapted enzymes include not only those with the optimal temperatures lower than the room temperature (generally 25 °C), like OalA from *Vibrio splendidus* 12B01 and the alginate lyase from *Pseudoalteromonas* sp. strain No. 272 whose optimum temperatures were 16 °C and 25 °C, respectively [21,70], but also the mesophilic ones with relatively high activities at the temperature ≤25 °C, even if their optimum temperatures are higher [33,71]. For example, as detailed in Table 3, the alginate lyases Algb and Alg7A from *Vibrio* sp. W13, AlgM4 from *V. weizhoudaoensis* M0101, and AlyS02 and Alyw201 from *Flavobacterium* sp. S02 and *Vibrio* sp. W2, respectively, reported by us previously all shared the optimal reaction temperature of 30 °C, but remained high activities at lower temperatures (Table 3), demonstrating decently cold-adapted properties [33,34,46,48,49]. Although recombinant Alyw208 had a little higher optimal temperature (35 °C), its relative activities at 25 °C, 20 °C, 15 °C, and 10 °C were 99%, 92%, 79%, and 71%, respectively (Figure 6A and Table 3). Therefore, this enzyme was more active at low temperatures than other cold-adapted alginate lyases. Moreover, when the temperature rose to above 55 °C, its relative activity remained as 73% of the maximum (Figure 6A), which was rare among cold-adapted enzymes, verifying a broader temperature range for the activity of Alyw208 with both cold-adapted and mesophilic characteristics than those of other enzymes of a sort.

The relative activity of recombinant Alyw208 was kept at around 87% after incubation for 1 h at 35 °C, but declined dramatically when the temperature rose to above 40 °C (Figure 6B). For the sake of improving thermostability, immobilization of enzymes is an effective means [72]. Li et al. immobilized the alginate lyase AlyPL6 onto mesoporous titanium oxide particles (MTOPs), and the immobilized enzyme presented excellent thermal stability and reusability [29]. However, recombinant Alyw208 maintained its thermostability at least below 35 °C, and this property satisfied its industrial utilization because this tool enzyme could run the catalytic processes at room or even lower temperatures without warming, which reduced the energy consumption, production costs, and contamination risks.

### 2.5. Effects of pH on the Activity and Stability of Recombinant Alyw208

The activity of Alyw208 was measured at different pH and the same temperature. As shown in Figure 7A, Alyw208 presented the highest activity at pH 10.0. After Alyw208 was incubated in buffer solutions of different pH, the remaining enzyme activity was measured. As shown in Figure 7B, the relative activity of Alyw208 remained at above 60% after the enzyme was incubated at pH 8.0–11.0 for 12 h at 4 °C. Therefore, Alyw208 is a high-alkaline and alkaline-stable enzyme.

As displayed in Table 2, a majority of alginate lyases maximize their biocatalytic activities at neutral to slightly alkaline conditions, while a few presented the highest activity in high-alkaline environments (pH ≥ 9.0) [33,35]. The extraction of alginate from brown seaweed generally needs high-alkaline conditions [73], and Alyw208 capable of efficiently degrading alginate polymers in high-alkaline environments could simplify the preparation of AOs into one step of simultaneous extraction and degradation. Moreover, since alginate dissolves preferably at high pH [73], the degradation of alginate at high concentrations by Alyw208 may improve the production efficiency of AOs. Most reported alginate lyases with optimal reaction pH higher than 9.0 were identified from marine molluscs or *Chlorella* viruses [41,74]. Alyw208 not only possessed the optimal performance at pH 10.0, but held above 70% of its maximum activity at pH 7.0–12.0. Notably, high-alkaline alginate-degrading enzymes derived from bacteria are rather rare. The optimal reaction pH of alginate lyase OUC-ScCD6 from *S. coelicolor* A3(2) was 9.0, which kept over 80% of catalytic function at pH 7–10 [64]. The alginate lyase AlyPL6 from *P. hainanensis* NJ-02 in the PL6 family exerted the maximum activity at pH 10.0, while its activity decreased dramatically out of the pH scope of 8.0–10.5 [29]. The pH properties of the above enzymes were obviously poorer than that of Alyw208. Another high-alkaline alginate lyase Alyw203, which was also identified from *Vibrio* sp. W2, owned similar pH property to Alyw208 with the highest activity at pH 10.0 and over 80% of the activity at pH 7–12, but regrettably, the highest activity appeared at 45 °C [50]. All the three alkaline alginate lyases mentioned above were not cold-adapted, and the only high-alkaline one with cold-adaptation except Alyw208 was A1m from *Agarivorans* sp. JAM-A1m, whose optimal pH and temperature were 10.0 and 30 °C, respectively [51]. However, when 200 mM NaCl was added to improve the enzyme activity, A1m showed the optimal performance at pH 9.0, and even under the optimum conditions, its activity was much poorer than that of the recombinant Alyw208 [51]. Interestingly, the alginate lyases with the best pH properties identified by now, Alyw208 and Alyw203, are both associated with the same bacterium *Vibrio* sp. W2 isolated from abalone viscera. Several reported alkaline alginate lyases come from mollusks, which indicates that the particular living environment in abalone viscera contributes to the outstanding pH properties of these two enzymes.

### 2.6. Effects of Metal Ions and NaCl on Alyw208 Activity

The impacts of different metal ions at 1 mM and 10 mM on Alyw208 were measured. As shown in Figure 8A, Na^+^ and Mn^2+^ at both concentrations increased the enzyme activity, while Al^3+^, Zn^2+^, and Co^2+^ inhibited the enzyme function. Ca^2+^ exerted a slightly inhibitory effect on Alyw208 (Figure 8A). For other ions, the cases were complicated. K^+^, Mg^2+^, and Ba^2+^ weakened the catalytic reaction at 1 mM, while they exerted improving effects at 10 mM (Figure 8A). On the contrary, Fe^3+^, Fe^2+^, and Cu^2+^ exhibited enhancing effects at 1 mM, but reduced the activity significantly at 10 mM (Figure 8A). With regard to the polysaccharide lyases from marine microorganisms, Na^+^ and K^+^ commonly enhance enzyme activities [47,75]. The positive influences of Ca^2+^ and Mg^2+^ and the negative impacts of Mn^2+^ on the activities of the majority of alginate lyases have been reported [19,49]. Surprisingly, a different phenomenon was found from Alyw208, which was dramatically stimulated by Mn^2+^ and slightly inhibited by Ca^2+^ (Figure 8A). Similar results were obtained from the alginate lyase Alg7A, whose activity was increased by Mn^2+^ as well [46]. Heavy metal ions, like Ba^2+^, Co^2+^, Zn^2+^, Cu^2+^, and Fe^3+^, generally decrease the biocatalytic competence [33], which may be due to the competitive binding of these cations against the substrates to the amino acid residues in the active center of the enzyme molecule [46]. Meanwhile, these cations can reduce the surface charge density and the ion interaction between the substrate and the enzyme molecule [76]. Nevertheless, there were exceptions. Co^2+^ acted positively on the alginate lyase Alyw201 from the same host as Alyw208 [48]. Similarly, Alyw208 and another alginate lyase aly-SJ02 from *Pseudoalteromonas* sp. SM0524 were stimulated by Ba^2+^ (Figure 8A) [77]. The diverse living environments of microorganisms give rise to different effects of ions on the enzymes originating from these strains. SDS slightly inhibited Alyw208 vigor, and favorable resistance to this surfactant was observed (Figure 8A). The chelating agent EDTA showed a significant inactivating effect on Alyw208 (Figure 8A), similar to most alginate lyases [31].

Since most alginate-lyase-producing microorganisms come from the ocean environment, Na^+^ influences the catalytic activities of alginate lyases. Therefore, we studied the effect of NaCl on the activity of Alyw208 and its tolerance to high NaCl concentrations. As shown in Figure 8B, Alyw208 activity increased as the NaCl concentration rose within the range of 0–3 M and peaked at 1.5 M NaCl, which was about 2.1 times the relative activity in the absence of NaCl. As the enzyme still displayed confirmative activity without NaCl (Figure 8B), it was salt-activated, but not dependent on Na^+^. Analogously, the activity of AlyS02 from *Flavobacterium* sp. S02 was boosted to around 250% by 500 mM NaCl [33]. The activation effects on other alginate lyases were even stronger. The activity of AlgM4 from *V. weizhoudaoensis* was increased to 7.41 times when 1 M NaCl existed [34]. The activity of A9mT from *Vibrio* sp. JAM-A9m was raised by 24 times by NaCl at an even lower concentration (400 mM) [61]. For the enzymes isolated from the marine environment, Na^+^ was believed to have a protection effect on the conformation of the active center at high temperatures [34,76]. However, the impact of NaCl on the recombinant Alyw208 was relative small, just like those on the other alginate lyases from *Vibrio* sp. W2. The activity of Alyw201 reached the highest of 124.7% in the presence of 0.75 M NaCl [48]. Alyw202 activity was almost doubled by 750 mM NaCl [52]. For Alyw203, the improving effect reached the strongest (relative activity of 148.2%) in the presence of 2 M NaCl [50]. Fortunately, these enzymes showed unimagined salt tolerance, unlike other alginate-degrading enzymes, whose activities declined dramatically when the concentration of NaCl changed to a little higher [33,34,44,49]. Due to the excellent tolerance to NaCl and ions, Alyw208 would be competent for industrial applications under different conditions.

### 2.7. Action Pattern and Final Products of Alyw208

In this work, several methods were employed to investigate the catalytic reaction pattern and the final products of recombinant Alyw208. Thin layer chromatography (TLC) was adopted to analyze the enzymatic degradation products, with the results illustrated in Figure 9A. During the degradation process, AOs were gradually generated and accumulated in polydispersity. Several apparent spots formed by the final products after 2 h incubation were intuitively observed on the plate, indicating that AOs with DPs 2–6 as well as alginate monomers were produced (Figure 9A). Then, the final products were detected using electrospray ionization mass spectrometry (ESI-MS) in the negative mode. As shown in Figure 9B, the ion peaks of [ΔDPx − H]^−^ (x = 1–6) appeared at 174.9, 351.0, 527.1, 703.1, 879.1, and 1055.0 *m*/*z*, respectively, and the ion peak at 439.2 *m*/*z* represented [ΔDP5 − 2H]^2–^ (Figure 9B). The result of ESI-MS was in line with that attained from TLC, which was similar to the consequence reported by Li et al. That is, the final catalytic products of the alginate lyase AlyPL6 from *P. hainanensis* NJ-02 contained sugars with DPs 1–4, and penta- and hexasaccharides accounted for a small fraction [29]. It is worth noting that some endo-type alginate lyases possess exolytic activity, which causes the existence of alginate monomers in the enzymatic products [41,78].

Endolytic alginate lyases show diverse product compositions (Table 2). The alginate lyases with single product distribution (one DP or two DPs) were considered conducive to further separation and purification of AOs [25,33,46,50,62]. However, oligomers with different DPs from enzymatic degradation of alginate manifested a series of physiological functions [43]. Alginate pentamers have an antitumor effect on osteosarcoma cells due to their anti-inflammatory and antioxidant activities [79]. Glyceollin could be induced by disaccharides ‘ΔG’ from enzymatically degraded alginate in soybean seeds [80]. Alginate oligomers with DPs 6–8 displayed a promising elicitor function in soybean cotyledon, improving plant defense against *P. aeruginosa* [81]. Preparing these different AOs through enzymatic degradation needs the participation of multiple endo-type alginate lyases with single product distributions. However, the use of Alyw208 alone can satisfy the demand for producing AOs with diverse DPs.

### 2.8. Preparation of AOs by Alyw208

Unsaturated oligosaccharides with double bonds at new non-reducing terminal uronic acids are born after the degradation of alginate by endo-type alginate lyases through the β-elimination mechanism [19]. The unsaturated AOs derived from enzyme catalysis possessed more desirable biological activities, which were primarily attributed to the double bonds in AOs [8,15]. Thus, AOs preparation has become an essential industrial application of alginate lyases.

We used Alyw208 to degrade 2% (*w*/*v*) sodium alginate solution (pH 10.0) at room temperature (20–25 °C), so as to examine the production of AOs. The viscosity of the sodium alginate substrate and the reducing sugar concentration in the substrate solution were determined at different time points. As shown in Figure 10, the viscosity of the sodium alginate solution declined rapidly at the initial stage of the reaction and was stable at a low level after 100 min, matching a typical endolytic fashion. After a continuous increase until 2 h, the concentration of reducing sugar in the sample reached 27.63 mM, then kept roughly steady, and finally reached 28.17 mM at 3 h (Figure 10).

The products after reaction for 2 h were further monitored by size-exclusion chromatography (SEC) through high-performance liquid chromatography (HPLC) (Appendix A). Along with the production of AOs, the conversion from unsaturated alginate monosaccharide to 4-deoxy-L-erythro-5-hexoseulose uronic acid (DEH) and subsequent hydration to 2, 4, 5, 6-tetrahydroxytetrahydro-2H-pyran-2-carboxylic acid (TPC) occurred spontaneously, with DEH and TPC having no absorbance at 235 nm [44]. Therefore, a refractive index detector was used for detecting the degradation products. Similar result to TLC and ESI-MS can be observed in Appendix A. That is, alginate monomer and AOs with DPs 2–6 were the main products, the proportions of which were calculated by peak area integration and presented in Appendix A. Tetramers accounted for the most part (28.62% for mass fraction), and other AOs also occupied significant proportions (16.47% for DP 2, 18.11% for DP 3, 21.91% for DP 5, and 9.75% for DP 6) (Appendix A). The proportions of monosaccharides cannot be calculated by peak area integration, perhaps due to their low content. Correspondingly, the concentration, yield (g_AOs_/g_sodium alginate_), and volumetric productivity of AOs with DPs 2–6 were 16.77 mg/mL, 0.95, and 0.14 mg·mL^−1^·min^−1^, respectively. The results proved that Alyw208, as a novel high-alkaline and cold-adapted alginate lyase, possessed a bright application prospect for the industrial production of AOs.

However, more studies should be performed in detail for the application of this enzyme in the future, including the optimization of the substrate concentration, the enzyme dosage, reaction temperature, time, and pH. In this work, 2% (*w*/*v*) sodium alginate and excess enzyme were adopted. In fact, the enzyme usage needs be reduced to control the cost. High substrate concentrations are conducive to improving the yield of AOs, but bring difficulties during degradation due to the high viscosity of this polysaccharide, so adding alginate substrate in batch will be a favorable method. In addition, the cold and high-alkaline adaptation endows Alyw208 with the capability to catalyze the preparation of AOs at room temperature in one step (simultaneous alkaline extraction and degradation of alginate), thus dramatically reducing the energy consumption and production costs and simplifying the preparation of AOs. In conclusion, efforts need to be made to obtain lower production costs and optimum productivity, so as to lay a foundation for industrial production of AOs by Alyw208.

## 3. Materials and Methods

### 3.1. Strains, Media, and Materials

Sodium alginate (M/G ratio: 1.66; viscosity: 100–200 cP) from *Macrocystis pyrifera* was purchased from Qingdao Bright Moon Seaweed Group Co., Ltd. (Qingdao, China). Homopolymeric alginates polyG (M/G ratio 1.8/98.2) and polyM (M/G ratio 97.3/2.7), as well as standard alginate mono- and oligosaccharides, were bought from Qingdao BZ Oligo Biotech Co., Ltd. (Qingdao, China). TLC silica gel 60 F254 plates were purchased from Merck KGaA (Darmstadt, Germany). *E. coli* DH5α used for plasmid construction was cultivated at 37 °C in the LB medium added with 50 μg/mL kanamycin if necessary. Prof. Zhen-Ming Chi from Ocean University of China kindly donated the pINA1312 expression vector and corresponding uracil mutant strain *Y. lipolytica* URA^−^ for gene expression. The transformants of *Y. lipolytica* were cultured on YNB plates prepared with 1.7 g/L yeast nitrogen base, 5.0 g/L (NH_4_)_2_SO_4_, 10.0 g/L glucose, and 25.0 g/L agar [48]. The GPPB medium used for recombinant protein expression was composed of 30.0 g/L glucose, 1.0 g/L (NH_4_)_2_SO_4_, 2.0 g/L yeast extract, 0.1 g/L MgSO_4_·7H_2_O, 3.0 g/L K_2_HPO_4_, and 2.0 g/L KH_2_PO_4_, and its pH was adjusted to 6.8 [48].

### 3.2. Bioinformatics Analysis of Alyw208

The online software SignalP 5.0 (http://www.cbs.dtu.dk/services/SignalP/, accessed on 14 August 2023) was employed to predict the cleavage position of the signal peptide. Conserved domains were predicted by CDD (https://www.ncbi.nlm.nih.gov/cdd). The online software pI/Mw Tool (https://web.expasy.org/compute_pi/) was employed to calculate theoretical Mw and pI. Based on other reported alginate lyases in the PL7 family, a bootstrapped phylogenetic tree was constructed using the neighbor-joining method in MEGA 7.0. The Vector-NTI program (Life Technologies, Grand Island, NY, USA) was used for the alignment of multiple protein sequences. The homology model of Alyw208 was built by the basic modeling module of MODELLER program (version 9.20) with the crystal structure of the PL7 alginate lyase AlyA from *Klebsiella pneumoniae* (PDB: 4OZX) as the template which has the highest homolog (identity of 61.35%) with Alyw208.

### 3.3. Expression and Purification of Recombinant Alyw208

The coding gene, *alyw208*, bearing a XPR2 signal peptide and a 6×His-tag at its 5′- and 3′-end, respectively, was synthesized by Synbio Technologies (Suzhou, China) after codon optimization and further ligated to the pINA1312 expression vector. The linearized fragment of *alyw208* was then transformed into *Y. lipolytica* URA^−^ according to the LiAc method described by Gietz and Schiestl [82]. After incubation in the GPPB medium for 84 h at 30 °C and 180 rpm, we detected a positive transformant which kept the highest extracellular alginate lyase activity.

The obtained supernatant following centrifugation at 10,000× *g* and 4 °C was loaded onto a His60 Ni Superflow column (Clontech Laboratories, TaKaRa, Dalian, China), which had been equilibrated by 50 mM Tris-HCl buffer (pH 9.0) with 300 mM NaCl, washed with the same buffer containing 20 mM imidazole, and then eluted with the same buffer at a linear gradient of imidazole (50–500 mM) by taking advantage of the affinity between 6×His-tag and Ni-IDA agarose [33]. The fractions presenting the alginate lyase activity, determined according to the description as follows, were pooled, desalted, and concentrated with a centrifugal filter 3 K device (3000 Da Mw limit) (Millipore, Burlington, MA, USA). The Mw of Alyw208 was assayed by SDS-PAGE on 12% (*w*/*v*) gel. BCA protein assay kits (Solarbio, Beijing, China) were wielded for examining the total protein concentrations in purified enzyme solutions.

The alginate lyase activity was determined following a method reported previously [48]. Briefly, appropriately diluted Alyw208 was mixed with 0.5% (*w*/*v*) sodium alginate solution in 50 mM Tris-HCl buffer (pH 10.0) at a ratio of 1:9 and incubated for 10 min at 35 °C. After the sample was boiled for 10 min to terminate the enzymatic reaction, the alginate lyase activity was assayed depending on the increase in OD_235nm_ originating from the unsaturated double bonds formed between C4 and C5 inside the uronic acid located at the new non-reducing end. One unit (U) of activity was defined as the amount of alginate lyase required to increase the absorbance by 0.1 per min.

### 3.4. Substrate Specificity and Kinetic Constants

To analyze the substrate specificity of recombinant Alyw208, we mixed 0.5% (*w*/*v*) polyG, polyM, and sodium alginate separately with 50 mM Tris-HCl buffer (pH 10.0) as substrate solutions, in which the enzyme activities were measured and compared under the standard conditions mentioned above.

The maximum reaction velocity (*V*_max_) and Michaelis constant (*K*_m_) of Alyw208 were determined by detecting the reaction velocity (V) of Alyw208 with the amount of reducing sugar generated per unit time at different substrate concentrations [S] (2.0–20.0 mg/mL sodium alginate) in the buffer of pH 10.0. The reciprocal 1/V of the reaction velocity was plotted against the reciprocal 1/[S] of the substrate concentration, and the *V*_max_ and *K*_m_ were calculated utilizing the slope and intercept [83]. The *k*_cat_ and *k*_cat_*/K*_m_ values were subsequently deduced.

### 3.5. Effects of Temperature and pH on the Activity and Stability of Alyw208

To investigate the optimum temperature of recombinant Alyw208, we determined the enzyme activity at 10–60 °C with the standard activity assay method described above. The relative activity was calculated, with the highest activity set as 100%. To assess the thermal stability, we incubated the alginate lyase at 10 to 60 °C for 1 h and measured the remaining activity, with the initial activity set as 100%. In addition, the optimal pH of Alyw208 was evaluated. Sodium alginate solutions were prepared with a series of 50 mM buffer solutions at pH 4.0–12.0 as the substrates, and the enzyme activity at the optimum pH was considered as 100%. For determining the pH stability of Alyw208, we incubated it at a series of pH and 4 °C for 12 h and examined its relative activity, with the initial activity regarded as 100%. All the reactions were repeated three times.

### 3.6. Effects of Ions and NaCl on Alyw208 Activity

For the influences of different metal ions and chemical reagents on recombinant Alyw208, the residual activity of Alyw208 was determined after 12 h incubation at 4 °C with the addition of Na^+^, K^+^, Mg^2+^, Fe^2+^, Fe^3+^, Ca^2+^, Cu^2+^, Zn^2+^, Al^3+^, Mn^2+^, Ba^2+^, Co^2+^, SDS, and EDTA at 1 mM and 10 mM. The mixture without any metal ion and chemical reagent was used as the control.

The activity of Alyw208 was measured at 35 °C using the substrates supplemented with 0–3 M NaCl. In this way, we studied the impacts of NaCl on the activity of the alginate lyase and its tolerance to high NaCl concentrations. The enzyme activity without the addition of NaCl was taken as control.

### 3.7. Final Product Analysis of Alyw208

To identify the degradation products of Alyw208, we added excess Alyw208 (50 U) to 10 mL of 2% (*w*/*v*) sodium alginate solution (pH 10.0) at room temperature (20–25 °C). The reducing sugar concentration was determined periodically with the DNS method until it no longer changed [33]. Simultaneously, an Ostwald viscometer (Shibata Scientific Technology LTD., Soka-City, Saitama, Japan) was used to determine the viscosity. Each sample was heated at 100 °C to terminate the catalytic reaction, and the reaction system was subsequently centrifuged for 15 min at 12,000× *g*. The supernatant was loaded on the centrifugal filter 3 K device (Millipore, Burlington, MA, USA) for simultaneous removal of macromolecules and concentration.

The depolymerization products at each time point during the reaction were identified by TLC according to the procedure described previously [33]. After filtration, the final product solution was mixed with methanol (1:1, *v*/*v*) and then directly injected into the ESI-MS instrument (Bruker Esquire HCT, Billerica, MA, USA). The mass-to-charge ratio (*m*/*z*) was profiled in the negative-ion mode under the conditions used previously [33]. In addition, the final product was loaded onto the Superdex^TM^ peptide 10/300 GL column (GE Healthcare, Boston, MA, USA) with 0.2 M NH_4_HCO_3_ as the eluent (0.6 mL/min) for gel filtration chromatography analysis on an Agilent 1260 Infinity HPLC platform equipped with a refractive index detector (Agilent Technologies, Santa Clara, CA, USA).

## 4. Conclusions

In this study, a new alginate lyase, Alyw208, of the PL7 family was identified from the marine bacterium *Vibrio* sp. W2 isolated from abalone viscera, and its expression and characterization were also conducted. The recombinant Alyw208 was cold-adapted and high-alkaline, with the optimal performance shown at 35 °C and pH 10.0, respectively. Compared to other alginate-degrading enzymes, it held a broad temperature range for the activity, maintaining the relative activity of over 70% at 10 to 55 °C. Moreover, the Alyw208 activity was improved by Na^+^ and increased to 2.1 times in the presence of 1.5 M NaCl. Finally, it was confirmed as an endo-type alginate lyase with the capability of degrading both polyG and polyM in spite of the polyG preference, producing alginate monosaccharides and oligomers of DPs 2–6 (DPs 1–6 in total). Therefore, Alyw208 has a promising application prospect in the biopreparation of AOs.

Hitherto, we have identified, expressed, and characterized four alginate lyases from *Vibrio* sp. W2 isolated from abalone viscera. Notably, all these enzymes possessed promising properties, i.e., the pH stability and cold adaptation of Alyw201, the high activity, ion tolerance, and pH stability of Alyw202, the high-alkaline adaptation, ion tolerance, and heat recovery of Alyw203, and the cold adaptation and high alkaline adaptation of Alyw208. Interestingly, these alginate lyases were all from the PL7 family and polyG-preferred. It seems that the four alginate lyases identified do not cover all the alginate-degrading enzymes the strain W2 contains and needs for thoroughly depolymerizing and utilizing alginate. Yagi et al. found through genome mining that eight alginate lyase-encoding genes (two from the PL6 family, three from the PL7 family, two from the PL17 family, and one from an unknown family) existed in the genome of *Shewanella* sp. YH1 [84]. The alginate lyases with diverse characteristics and alginate-degrading ability from different PL families in *Sphingomonas* sp. A1 (such as A1-I, A1-II, A1-III, A1-IV, A1-II’, and A1-IV’) co-conducted the exhaustive degradation of alginate [66]. Similarly, there should be other alginate lyases that can be identified in *Vibrio* sp. W2 as well. We will focus on sufficient mining of the genome of W2 strain to characterize all its alginate-degrading enzymes so that the entire alginate catabolic mechanism can be elucidated in detail.

## Figures and Tables

**Figure 1 molecules-28-06190-f001:**
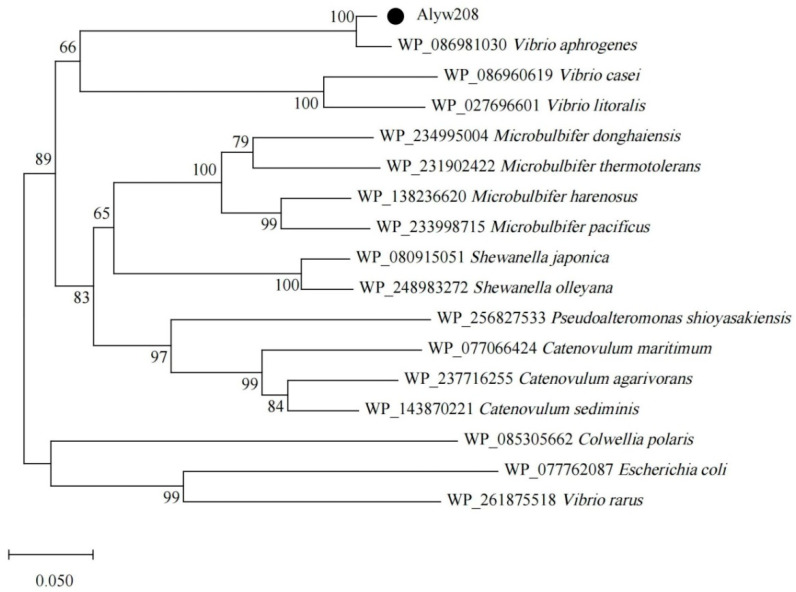
Phylogenetic tree of Alyw208 and related alginate lyases in the PL7 family. The black circle indicates the alginate lyase in this work.

**Figure 2 molecules-28-06190-f002:**
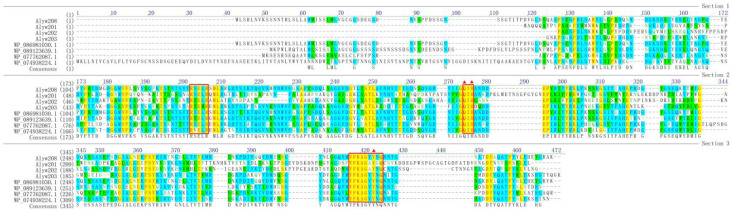
Comparison of amino acid sequences between Alyw208 and other alginate lyases in the PL7 family, including Alyw201 (Genbank number MT232847), Alyw202 (Genbank number MT424751), Alyw203 (Genbank number OR423046), the alginate lyase from *Vibrio aphrogenes* (Genbank number WP_086981030), the alginate lyase from *Vibrio algivorus* (Genbank number WP_089123639), the alginate lyase from *Escherichia coli* (Genbank number WP_077762087), and the alginate lyase from *Algibacter lectus* (Genbank number WP_074938224). The typical conserved regions of PL7 family are boxed in red. The red triangles highlight the potential residues involved in the catalytic activity of Alyw208.

**Figure 3 molecules-28-06190-f003:**
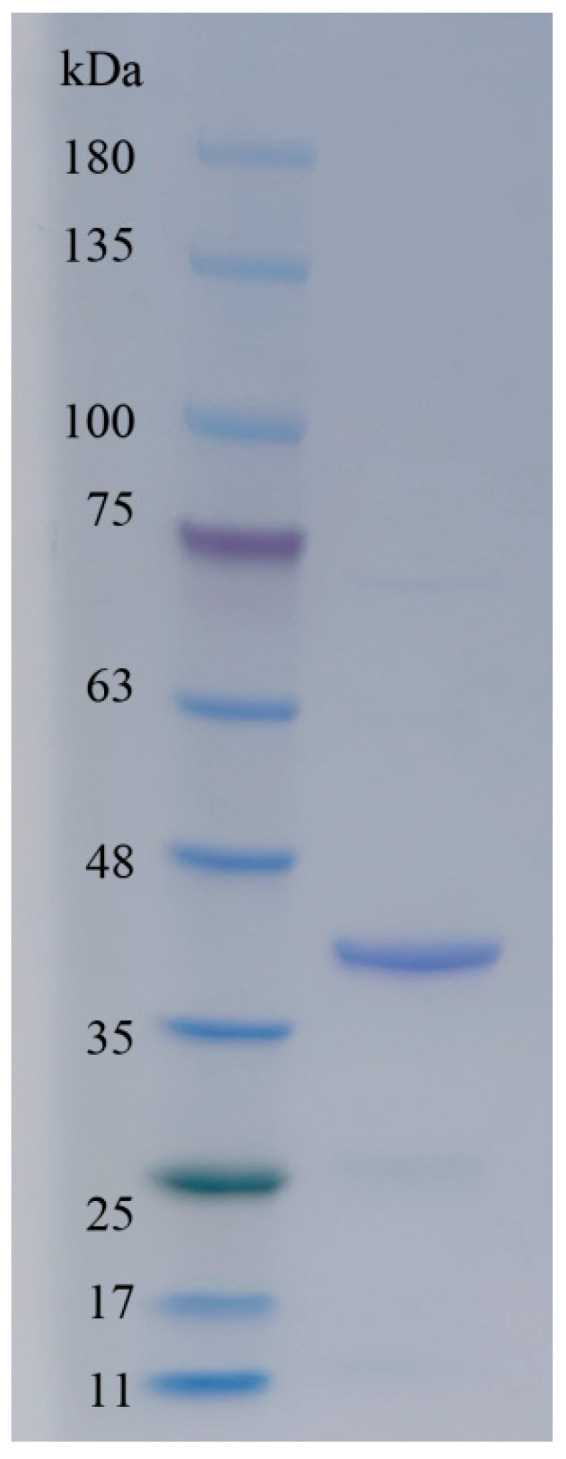
SDS-PAGE of purified Alyw208. The left lane displays the results of protein markers, whose Mws were noted on the left of the lane; the right lane corresponds to recombinant Alyw208.

**Figure 4 molecules-28-06190-f004:**
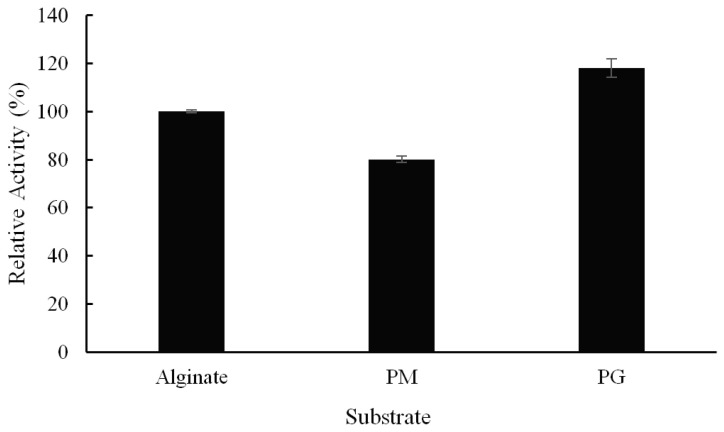
Substrate specificity of recombinant Alyw208. Alginate: sodium alginate (M/G ratio: 1.66); PM: polyM (M/G ratio 97.3/2.7); PG: polyG (M/G ratio 1.8/98.2). The relative activity toward alginate was considered as 100%.

**Figure 5 molecules-28-06190-f005:**
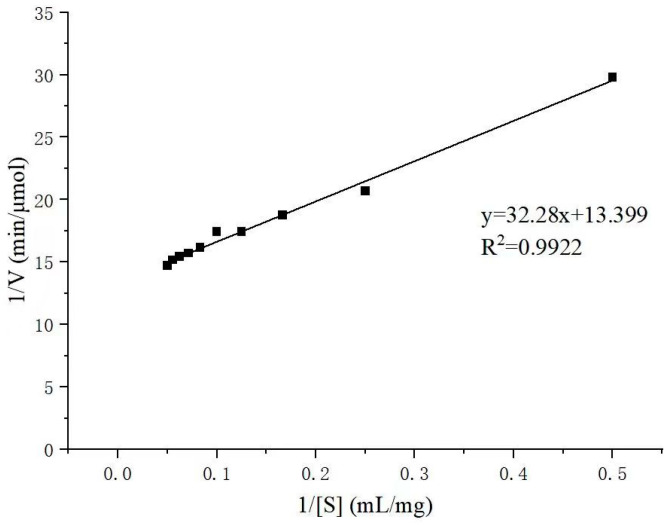
Lineweaver–Burk plot of alginate degradation by Alyw208.

**Figure 6 molecules-28-06190-f006:**
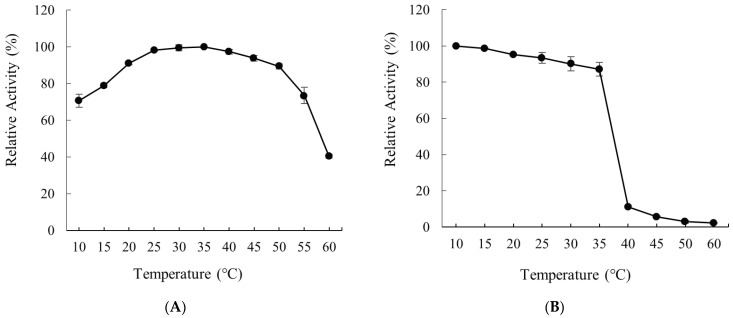
(**A**) Effect of temperature on the activity of Alyw208, with the highest activity set as 100%. (**B**) Effect of temperature on the stability of Alyw208. The alginate lyase was incubated at 10 to 60 °C for 1 h and the relative activity was measured by the standard activity assay, with the initial activity set as 100%.

**Figure 7 molecules-28-06190-f007:**
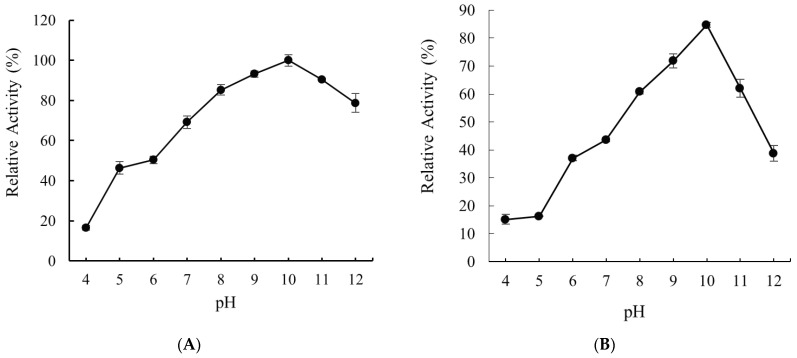
(**A**) Effect of pH on the activity of recombinant Alyw208, with the highest enzyme activity set as 100%. (**B**) Effect of different pH on the stability of Alyw208. The enzyme was incubated at 4 °C for 12 h at a series of pH, and the relative activity was determined, with the initial activity set as 100%.

**Figure 8 molecules-28-06190-f008:**
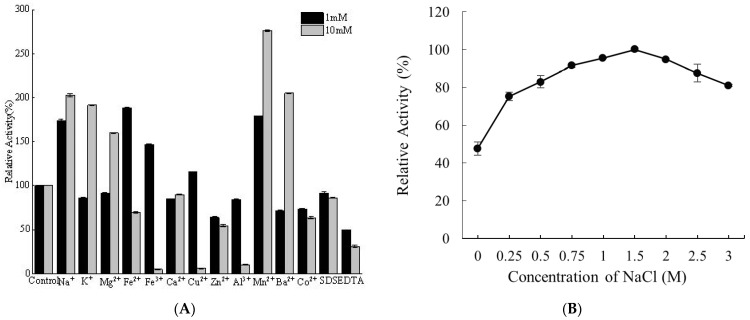
Influences of different metal ions and chemical reagents on recombinant Alyw208. (**A**) Effects of metal ions, SDS, and EDTA on the activity of Alyw208. (**B**) Effect of NaCl on the activity of Alyw208. Here, the maximum activity was set as 100%.

**Figure 9 molecules-28-06190-f009:**
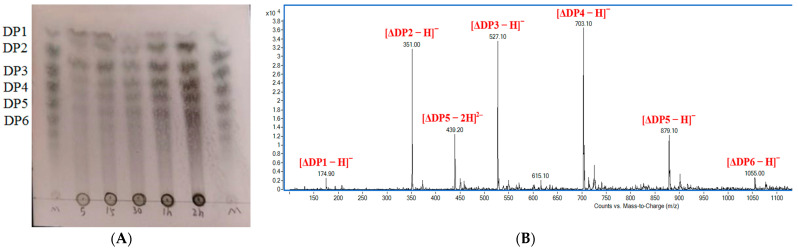
Analysis of degradation products by TLC and ESI-MS. (**A**) TLC analysis of products at a series of reaction time points. Lane M shows a mixture of alginate monomers (DP 1) and different oligomers (DPs 2–6); lanes marked with 5, 15, 30, 1 h, and 2 h represent the samples collected at the reaction time points of 5 min, 15 min, 30 min, 1 h, and 2 h, respectively. (**B**) ESI-MS analysis of final degradation products.

**Figure 10 molecules-28-06190-f010:**
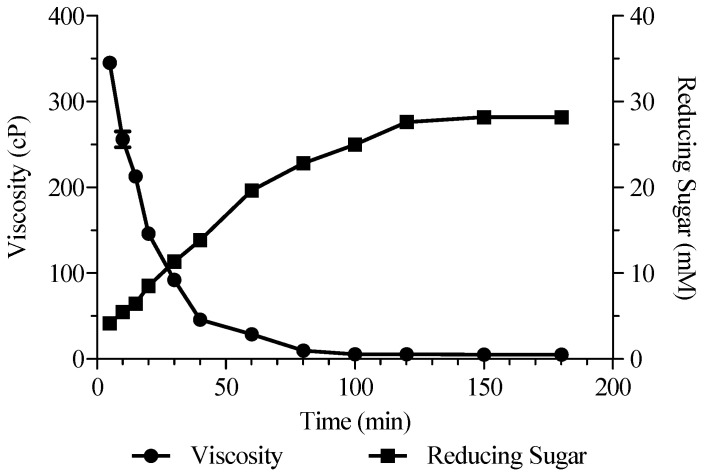
Viscosity and reducing sugar content during enzymatic reaction, where 2% (*w*/*v*) sodium alginate was degraded by Alyw208 at room temperature.

**Table 1 molecules-28-06190-t001:** Summary of the purification of Alyw208.

Purification Step	Total Protein (mg)	Total Activity (U)	Specific Activity (U/mg)	Purification Fold	Yield (%)
Crude enzyme	32.23 ± 2.59	3776 ± 305	117.2	1	100
Ni-IDA agarose	3.42 ± 0.25	2495 ± 234	729.5	6.22	66.1

**Table 2 molecules-28-06190-t002:** Properties of Alyw208 and other alginate lyases.

Enzyme	Source	PL Family	Optimal Temperature (°C)	Optimal pH	Substrate Specificity	Products (DP)	Reference
Alyw208	*Vibrio* sp. W2	7	35	10.0	polyG, polyM	1–6	This study
Alyw201	*Vibrio* sp. W2	7	30	8.0	polyG, polyM	2–6	[48]
Alyw202	*Vibrio* sp. W2	7	45	9.0	polyG, polyM	2–4	[52]
Alyw203	*Vibrio* sp. W2	7	45	10.0	−	1–2	[50]
Algb	*Vibrio* sp. W13	7	30	8.0	polyMG, polyG, polyM	2–5	[49]
Alg7A	*Vibrio* sp. W13	7	30	7.0	polyG, polyM, polyMG	3	[46]
AlySY08	*Vibrio* sp. SY08	−	40	7.6	polyG, polyM	2	[25]
Aly510-64	*Vibrio* sp. 510-64	−	35	7.5	polyG, polyMG	3	[63]
AlgM4	*Vibrio weizhoudaoensis* M0101	7	30	8.5	polyG, polyM	2–9	[34]
AlyS02	*Flavobacterium* sp. S02	7	30	7.6	polyG, polyM	2, 3	[33]
Alg2A	*Flavobacterium* sp. S20	7	45	8.5	polyG	5–7	[60]
PA1167	*Pseudomonas aeruginosa* PAO1	7	40	8.5	polyMG	2–4	[32]
AlgA	*Pseudomonas* sp. E03	5	30	8.0	polyM	2–5	[30]
OUC-ScCD6	*Streptomyces coelicolor* A3(2)	6	50	9.0	polyM, polyG	2–6	[64]
TsAly6A	*Thalassomonas* sp. LD5	6	35	8.0	polyG, polyM	2, 3	[65]
AlyPL6	*Pedobacter hainanensis* NJ-02	6	45	10.0	polyMG, polyG, polyM	1–4	[29]
A1-IV’	*Sphingomonas* sp. A1	15	50	8.5	polyM, polyMG	2, 3	[66]

**Table 3 molecules-28-06190-t003:** Relative activities of Alyw208 and other reported cold-adapted alginate lyases at lower temperatures.

Alginate Lyase/Source	Relative Activity at 25/20/15/10 °C (%)	Optimal Temperature (°C)	Reference
Algb/*Vibrio* sp. W13	−/75/−/50	30	[49]
Alg7A/*Vibrio* sp. W13	−/90/−/70	30	[46]
AlgM4/*V. weizhoudaoensis* M0101	−/90/−/50	30	[34]
AlyS02/*Flavobacterium* sp. S02	90/70/50/40	30	[33]
Alyw201/*Vibrio* sp. W2	90/80/50/40	30	[48]
Alyw208/*Vibrio* sp. W2	99/92/79/71	35	This study

## Data Availability

All the data of this study are available within the paper and its Appendix A file.

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
