# Peer review of "A Novel Cold-Adapted and High-Alkaline Alginate Lyase with Potential for Alginate Oligosaccharides Preparation"

_molecules, 2023, doi:10.3390/molecules28176190_

Round 1

Reviewer 1 Report

The manuscript describe an alginate lyase Alyw208 Vibrio sp. W2, was expressed in Yarrowia lipolytica of food grade, the enzymatic characteristics of the enzyme was determined, and Alyw208 was suggested to be a potent candidate for the industrial applications. No major issues was pointed and some minor issues should be corrected before accepted.

1.       The accession No. of Alyw201-203 should be provided in the Figure legends of Figure 1

2.       The error bar of Summary of the purification of Alyw208 in Table 1 should be provided.

3.       The Km and Vmax should be written in italic in Line208-209.

4.       The resolution of Figure 9. B was not clear.

Reviewer 2 Report

It is a very nice paper where alginate lyase has been purified to homogeneity level and extensively characterized. However, the manuscript can be further improved by the revisions suggested below.

 1.       Give the chemical equation catalyzed by this specific enzyme and the EC number of the specific alginate lyase studied here.

2.       Mention in which specific industries, highly alkaline/cold-adapted alginate lyase can be used, especially with high alkaline activity.

3.       Retrieve the 3D structure from the Alphafold database or generate the structure and include it in the main text with catalytic residues highlighted.

4.       The purification of the enzyme is nice, so determine kcat from Vmax and compare kcat, Km and kcat/Km with other from similar organisms.

5.       Calculate activation energy from the data in figure 6a. From kcat and activation energies determine thermodynamic activation parameters of the enzyme reaction and comment. See Jayawardena, et al. 2017 Process Biochemistry, Volume 57, Pages 131-140).

6.        Lines 353-360: Figure S1 is a very important result. This figure should be in the main manuscript and not in the supplementary. Lines 353-360 should be under a separate heading entitled “Productivity analysis”. Expand the section in view of the recent review (Evaluating Enzymatic Productivity—The Missing Link to Enzyme Utility. Int. J. Mol. Sci. 202223, 6908. https://doi.org/10.3390/ijms23136908). Calculate volumetric productivity/specific volumetric productivity units as described in the reference. If productivity data is available from other studies on the same enzyme, compare it in the paper. Also, explain why viscosity is zero at 90 min but reducing sugars kept forming until 120 min?

needs polishing.

Round 2

Reviewer 2 Report

Figure 5, the 1/[S] axis should be extended beyond 0 to minus.

Proof reading may be needed.

Author Response

Response to Reviewer 2 Comments - Round 2

Figure 5, the 1/[S] axis should be extended beyond 0 to minus.

Authors’ response:

We would like to appreciate the recommendation. The Figure 5 has been replotted, and the abscissa axis has been extended.